# Study on the Performance of Recycled Coarse and Fine Aggregates as Microbial Carriers Applied to Self-Healing Concrete

**DOI:** 10.3390/ma16062371

**Published:** 2023-03-16

**Authors:** Zhenhua Duan, Zhenyuan Lv, Jianzhuang Xiao, Chao Liu, Xiangyun Nong

**Affiliations:** 1Department of Structural Engineering, College of Civil Engineering, Tongji University, Shanghai 200092, China; 2College of Science, Xi’an University of Architecture and Technology, Xi’an 710055, China

**Keywords:** recycled coarse aggregate, recycled fine aggregate, self-healing concrete, mechanical properties, analytical modeling

## Abstract

The contradiction between the scarcity of natural resources and the demand for construction materials has given rise to the application of recycled aggregates. Microbial self-healing concrete (SHC) is a clean and smart material, and its carrier has a great influence on repair performance. In this paper, recycled coarse aggregate (RCA) and recycled fine aggregate (RFA) were used as carriers, and their different repair effects over time were intensively investigated. The results showed that the RCA carrier had a better repair effect compared with that of RFA, and the maximum healing width could reach 0.27 mm by 28 day. The microbial repair efficiency was significantly influenced by the distribution of old mortar, with the RFA specimen having a small volume and wide distribution of repair products, while the RCA repair showed a centralized tendency. In addition, SEM, MIP and XRD characterization were used to analyze the repair mechanism. The time-dependent repair model was developed, and the applicability of the model for concrete enhancement under microbial repair was verified through experimental results. The research results could promote industrial applications by giving intelligent and green properties to recycled aggregates.

## 1. Introduction

Concrete, as the most consumed building material, has been in constant demand along with the development of the global construction industry since the invention of cementitious materials. Natural sand and gravel originate from environmental mineral deposition weathering and are important components that make up the concrete skeletal system [1]. The overuse of concrete materials has led to a scarcity of the remaining quantity, which is obviously contrary to the current progress of green and sustainable development of buildings. In addition, cementitious materials are subject to diversified degrees of cracking influenced by the hydration level, preparation process and maintenance environment. These microcracks reduce the strength and durability of concrete structures [2]. Concrete internal rebar faces the threat of de-passivation, and the rusted state weakens the integrity of the reinforcement concrete, causing structural failure due to the appearance of penetration cracks. In order to reduce the harm caused by the cracking, traditional remedial methods are used that mostly consist of external mortar-filling of concrete surface micro-cracks. Yet, the internal cracks and damages are not healed by this method. This method can only maintain the external cracks for a short period of time until they are disrupted again due to the presence of internal inherent cracks [3]. Therefore, the traditional repair effect is limited and high-cost, so it cannot effectively solve the problem. As an alternative, higher-quality NRM and concrete design ratios have been persistently developed to reduce the impact of cracking on overall strength reduction [4]. However, this usually means more cement consumption and extraction of NRM, which is not conducive to green and sustainable development in the construction industry.

Based on the premise of maintaining structural safety, the resource recovery of C&DW has become a hot topic of concern in the industry, mainly in the exploration of reducing natural material consumption [5,6]. Granular and powder products are obtained through the process of cutting, crushing and screening primary C&DW. As the main product derived from this process, RAs’ replacement of NS&G to prepare RAC has become a way to resolve the scarcity of natural resources [7]. According to their physical properties, RCA (particle size greater than 4.75 mm, Recycled Coarse Aggregate for Concrete GB/T 25177-2010) is used to replace natural coarse aggregates, and RFA (particle size less than 4.75 mm, Recycled Fine Aggregate for Concrete and Mortar GB/T 25176-2010) replaces natural sand. The development and application of RCA are in relatively early stages, and the performance of RCA concrete has been explored by scholars. Studies have shown that the RCA particle size, the amount of old mortar and the replacement ratio all affect the concrete mixes’ properties [8]. The mechanical properties of RCA concrete are mainly influenced by the aggregate state [3], type [9], particle size [10], replacement ratio [11], and fatigue life of the concrete [12]. Lower aggregate properties and higher replacement ratios lead to lower compressive, tensile and flexural strengths, and modulus of elasticity of the concrete [2]. The properties of RAC, such as carbonation [13] and freeze–thaw [14], deteriorate due to the incorporation of RCA compared to natural concrete. The more RCA is mixed and the lower the strength of the old mortar, the greater the dry shrinkage of the concrete within a certain dry shrinkage age for the same mix ratio [15,16]. With the surge in demand for natural sand, research on the performance of RFAs for concrete preparation has also emerged. However, it was found that the higher water absorption and poorer performance of RFAs made them less applicable than RCAs [17]. Their negative effects on concrete strength [18,19], shrinkage [20] and durability performance [21,22] are more significant at high substitution rates due to the increase in microcracks and porosity. Therefore, the higher concrete cracking and porosity degree due to the macro performance loss caused by the replacement of RCA and RFA is the main reason to limit the high added value of recycled products [23].

For concrete cracking, which is a time-dependent process, internal and external fine cracks cannot be detected in time by visual inspections, which results in a serious lag in manual repair [24,25,26]. Self-healing technology [27,28], as an intelligent phase change regulation mode, is suitable for the repair of materials due to its characteristics of “autonomous excitation, passive repair”. The suitability of polymeric materials [29,30], shape-memory alloys [31,32] and microorganisms [33,34,35] for concrete crack repair has been investigated by many scholars. The polymer material is pre-injected with microcapsules and hollow fiber tubes embedded in concrete to achieve the repair effect. The dry shrinkage and cracking of the concrete cause the thin wall of the package to break, and the polymer fills and hardens along the microcracks, providing good bonding and strength at the cracked interface [36]. Yet, the high cost and casting process requirements lead to poor engineering adaptability [37]. Shape-memory alloys are materials composed of two or more metallic elements that have a shape-memory effect [38]. The material deformation characteristics are the result of the reversible transformation of the low-temperature phase (martensite) to the high-temperature phase (austenite) when heated by the thermoelastic martensitic phase transformation, which enables crack and defect repair by alloy deformation inside the concrete [39]. Moreover, additional repair energy consumption and alloy location differences result in a large dispersion of healing effectiveness. Bioconcrete involves the repair of micro-cracks by the production of mineral materials through microbial activity in the concrete [40]. This process reduces concrete cracks to improve the strength and durability of the structure and addresses the structural maintenance needs of reinforced concrete. Microbial mineralization highlights the green properties of concrete more than other self-healing methods, as it is a natural process and environmentally friendly [41]. The self-healing process is directly related to calcium carbonate (CaCO_3_) production, which depends on many factors, including the concrete pH, microbial carriers and the amount of Ca^2+^ (free-state) in the concrete [42,43]. In addition, the type and concentration of bacteria, the type of carrier and the method of incorporation also affect the effectiveness of healing [44]. *Bacillus pasteurii*, a kind of self-healing bacteria, have a high alkaline activity and oxygen sensitivity, which are compatible with the properties of concrete materials.

Unlike the previous study [45], which considered the effect of microbial carriers of only RCA on the performance of concrete, this paper investigates the properties of RCA and RFA as carriers acting together in concrete (Figure 1). More importantly, the improvement in the performance of SHC due to differences in carriers was evaluated. By using an orthogonal experimental design, the effects of the carrier particle size, bacterial solution concentration and carrier replacement ratio on the repair efficiency were evaluated. The optimal solution for repair efficiency was also proposed. The time-dependent repair model at the crack was developed, and the effect of the crack repair trend at the interface with the carrier was summarized. The usefulness of the model was verified by designing the permeability test. The higher water absorption of RFAs results in a typically lower replacement ratio than RCAs when concrete performance was previously considered. Being a self-healing carrier provides a new avenue for high value-added applications, and RAs serve as a means of enhancing smart materials, playing a “turn waste into treasure” role.

## 2. Materials and Self-Healing Principles

### 2.1. Microbial Materials for Repair

The microbial strains used in this experiment were aerobic bacteria (Latin name is *Bacillus pasteurii*), and the culture site of the strain is Xi’an Institute of Microbiology. The specific culture method was used to inoculate the lyophilized strain in liquid medium. According to the national standard (‘Microbiological examination of food hygiene–Examination of Bacillus cereus’ (GB/T 4789.14)) for proliferation and testing, the mixed bacteria were diluted to an optical density state of 1.2 (the density of optical energy absorbed by object, the unit of detection is OD, OD = lg (1/trans)). The concentration of bacteria in the solution was about 2.4 × 10^9^ cfu/mL. The composition of the growth medium used was: ultrapure water (UPW), peptone, meat extracts, agar, NaHCO_3_ and Na_2_CO_3_. The lyophilized strain used for activation on slant tubes was placed in 2–8 °C refrigeration (vacuum freeze-drying method), and the content of components is shown in Table 1. After inoculation into the liquid medium, the cultures were incubated in the incubator shaker at 30 °C and 120 r/min for 24 h and then placed in the biochemical incubator at 30 °C.

### 2.2. Recycled Aggregate Carrier

The RCA and RFA for the test were provided by a Shaanxi recycling plant and were taken from demolished ring beams and structural columns in Xi’an city. The appearance of the RA was found to be flat, with prominent angles and wrapped in a layer of cement mortar, so its surface was rough and had voids. The RA was divided into RCA (particle size range 5–20 mm) and RFA (particle size range 1.18–4.75 mm) through manual sieving, where the fineness modulus of RFA was 3.05. Based on “the pebble and crushed stone for construction” (GB/T 14685) and mercury intrusion porosimetry (MIP) to quantify the basic differences in RA, the physical property indexes are shown in Table 2.

### 2.3. Adhered Old Mortar

The old mortar (OM) is the microbial sojourn area where repair behaviors are stimulated. To investigate the properties of the OM after service, fresh mortar was selected for comparison. The OM was hand-crushed from RCA, and similarly, the new mortar (NM) was crushed from concrete specimens and hydrated and cured for 28 day at the same mix ratio as the OM. By quantifying the difference in pore size between OM and NM by MIP, the data showed that the pore size of the adhered OM was larger than that of the NM. The cumulative pore volume increased more rapidly at the beginning of the mercury compression process, and its pore size could reach 28.63%. Due to the small pore size of the NM, there was no obvious trend in the growth of its accumulated pore volume at the beginning of the mercury compression process, and its porosity reached 17.68% (only 61.75% of that of the OM). In addition, the maximum pore volume accumulation value of the NM was 0.0834 mL/g, which is only 55.3% of that of the OM (0.1507 mL/g). With the increase in mercury accumulation, the larger pore size of the OM made rapid growth, which then slowed down and grew again. However, the small pore size of the NM showed a tendency to increase significantly lagging growth. The cumulative pore volume and pore size trends of the OM and NM are shown in Figure 1.

The pore size of the OM has peaks at points 1, 2 and 3, as shown in Figure 2. Point 3 is the maximum value measured by the mercury-pressure intrusion process, which indicates that the OM pore size has the best pore size connectivity in the range of 0.5 × 10^6^–1 × 10^6^ nm. The NM has peak pore size values at points 1 and 2, with values in the range of 10–100 nm. This indicates that the NM pore-size connectivity is mostly smaller and its maximum pore-size connectivity is also lower than that of the OM. Among them, the pore maximum values of the OM (point 3) are 0.008 mL/g and 0.031 mL/g higher than the other two peak points, respectively, which indicates that the pore size difference in the old mortar has a large range of fluctuating values. This is characterized by its loose structure and inhomogeneous pores. The difference between the maximum pore size of the NM (point 2) and the other pore size values is only 0.009 mL/g, which indicates that the NM is more compact. Additionally, the maximum pore size is also much smaller than the maximum pore size of the OM.

With the increase in initial pressure, the increment of invasion of OM is more obvious than that of NM, which means that not only the maximum diameter but also the connectivity of OM pores also greatly exceeds that of NM. As shown in Figure 3a, the early intrusion increment limit point 1 of the OM takes a value of 11.41%, which is about 232% of the NM result (4.91%); at subsequent pressure increments, the cumulative porosity varies in the range of 0.22% to 1.85%. However, at the end of the mercury compression process, the intrusion increment of the OM is significantly lower compared to the NM, which also indicates that the connectivity of the OM in the micropores is much lower than that of the NM. The intrusion increment limit point 3 of the OM takes a value of 4.10%, which is about 55.7% of the peak intrusion increment limit point 4 of the NM (7.18%), as shown in Figure 3b.

### 2.4. Other Materials

The cement used in the SHC was PO 42.5, and its chemical composition is listed in Table 3. The natural coarse aggregate consisted of continuously graded aggregates of 5–20 mm in size. The natural sand had a fineness modulus of 3.1, and the mixing water and additional water for the specimens were tap water.

### 2.5. Self-Healing Principle

The dormant state of *Bacillus pasteurii* placed in the pores of the mortar attached to the RA was stimulated to awaken by water molecules and O_2_ from the air in the cracks. The microbial spores attract free-state Ca^2+^ and the Ca^2+^ attaches to the spore and combines with free-state CO_3_^2−^ to form CaCO_3_ (calcite state), as shown in Figure 4. The distribution of CaCO_3_ is transferred from the carrier inside the structure to the outer edge of the fracture. However, the particle size and distribution on the surface are more prominent compared with the crack interior, due to the superior moisture and oxygen conditions in the fracture area. The amount of Ca^2+^ within the concrete decreases with the formation of CaCO_3_. Calcium lactate as a bacterial nutrient effectively replenishes the calcium source while avoiding the weakening of the passivation effect of Ca(OH)_2_ on the reinforcement.

SHC is affected by its own hydration, dry shrinkage and external environment, so microcracks will appear in the protective layer and extend to the inside. Airborne ions will move from the OM to the RA inside the structure through the crack transmission channel, where harmful ions will have a depassivating and rusting effect on the reinforcement. Microorganisms within the adhering mortar will automatically stimulate repair behavior with the appearance of cracks. With their growth over time, the surface area will gradually repair from the bed points of the bilateral edges to the center until the cracks have completely disappeared, as shown in Figure 5.

## 3. Experimental Design and Methods

### 3.1. Carrier Preparation

The preparation of RCA and RFA as the carriers includes two stages: microbial adsorption and aggregate drying. The microbial adsorption was performed using a negative pressure vacuum adsorption method, i.e., the RA and bacterial solution were placed in a sealed glass container, and the vacuum pump was set at −0.6 MPa for 2 min, as shown in Figure 6. It should be affirmed that when the bacterial solution failed to completely wrap the RA, the carrier needed to be rotated for 30 s and then adsorbed again under negative pressure. After adsorption, the aggregate was dried in a constant temperature oven at 40 °C for 24 h. *Bacillus pasteurii* was made dormant to avoid the influence of strong alkalinity on its activity during mixing, and the water in the bacterial solution was avoided to prevent it from affecting the water–ash ratio of the specimen, as shown in Figure 7.

### 3.2. Mixing RATIO Design

Based on the previous experimental work [38], 54 standard specimens of 40 × 40 × 160 mm^3^ (30% carrier replacement ratio, 5–10 mm RCA particle size, 1.18–4.75 mm RFA particle size, 40% concentration of the bacterial solution, and 2 mm precast crack width) were prepared. This design is the result of a combination of concrete performance, crack repair efficiency, and microbial cost factors preferentially considered. The high replacement ratio of recycled materials reduces the basic performance of concrete and cannot be compensated by repair. In addition, it implies more microbial applications, which are clearly uneconomical. Moreover, the average living space of individual strains is reduced, and the activity is instead reduced at high concentrations. Among them, the SCC group was 18 SHC with RCA as the microbial carrier, the SFC group was 18 SHC with RFA as the microbial carrier, the SC group was 9 SHC with the bacterial solution directly mixed in during mixing, and the NC group was 9 control specimens prepared with NA. The carrier samples were pre-absorbed with a concentration of 40% bacterial solution, and calcium lactate was added as a nutrient after the microorganisms had awakened at the time of concrete mixing (Table 4).

### 3.3. Pre-Set Cracks and Curing

The specimen was loaded via the electro-hydraulic servo universal testing machine (GB/T 16826) using the three-point method to pre-set the crack. The specific method was to commission the press to load at a rate of 0.05 mm/min and to stop loading when a 0.1–0.3 mm crack appeared at the tensile side of the specimen and hold the load for 90 s. A crack observation point was set at 10 mm along the crack to record the initial crack width with a concrete crack width observation instrument. It should be emphasized that, in order to ensure the integrity of the specimen, the crack extension length should not exceed 30 mm, so there are only three measurement values for each specimen. The initial crack widths of the specimen marker points and their calculated values are shown in Table 5. In order to better match the actual engineering crack repair environment, the specimens were placed in a natural environment with water maintenance.

### 3.4. Repair Characterization

The apparent characterization of repair effectiveness over time was recorded by the crack observer with an accuracy of 0.05 mm to record the crack changes. By performing crack observation point width displacement measurements at 7 day, 14 day and 28 day, the average values of the prefabricated crack control initial widths of different specimens were obtained and are listed in Table 5. The crack marker points were characterized by the point repair rate to characterize the local repair efficiency (Equation (1)); the specimens were characterized by the isometric area repair rate to characterize the overall repair efficiency (Equation (2)).
(1)$$w_{p}=\frac{d_{0}-d_{t}}{d_{0}}\times 100\%,$$

*w_p_*—the repair ratio of the marker point.

*d*_0_—the initial crack width at the marker point.

*d_t_*—the remaining crack width at the marker point after repairing *T* days.
(2)$$w_{a}=\frac{\sum_{i=0}^{n}d_{i}\times l_{r}}{d_{0}\times l_{0}}\times 100\%,$$

*w_a_*—the repair ratio of the area (the width of the area between marker points is assumed to be equal).

*d_i_*—the effective repair width between marker points.

*l_r_*—the corresponding length of the effective repair width.

*d*_0_—the initial crack width of the repaired area.

*l*_0_—the corresponding length of the initial crack width.

The specimens were characterized by the repair carrier interface, total amount and morphology of repair products and product type. Among them, a ZEISS Sigma 500 integrated spectral scanning electron microscope (ZEISS Sigma, Oberkochen, Germany) was used for SEM characterization of the crack repair surface. The system was used non-destructively to detect the total amount of repaired material in the cracked area.

### 3.5. Water Permeability Test of Cracks

Drawing on previous experimental analysis of optimal solutions for RA carrier repair efficiency [38], an orthogonal design test was designed to determine the water flux at 14 day for samples with different crack repair rates. The initial crack water flux value *q*_0_ and the value *q_t_* for a repair time of *T* days were recorded for each group of specimens. As shown in Figure 8, the measurement required wrapping both sides of the crack and the lowest part of the notch with sealed waterproof tape. The length and width of the notch were 5 mm and 0.05–0.1 mm, respectively. After the control valve was used to adjust the flow rate, the specimen holder was placed on the upper side of the measuring cup and the inlet pipe was filled with 500 mL of water (set to a constant water pressure). The water storage tank, outlet pipe and crack surface were connected, and the concrete–acrylic box interface was coated with epoxy resin in a circular fashion. Transient values were recorded 5 min after the epoxy resin had solidified. After half an hour, the single maximum value was compared with one-sixth of the total water loss. The maximum value was taken as the actual water flux and compared between all sets of parameters. It should be stated that, since the water penetration method cannot determine fully repaired cracks, the measurement data are only valid for incompletely healed specimens.

## 4. Time-Dependent Model for SHC after Cracking

### 4.1. Optimization of Time-Dependent Characterization Methods for Repair Efficiency

The time-dependent measurement of repair performance was based on the standard rectangular crack assumption, and the repair trend was characterized by the change in the area of the marked points, as shown in Figure 9. In fact, all cracks showed an irregular, jagged development, which cannot be accurately characterized by the area method. In addition, a limitation of the area method is that only the surface repair product volume can be considered, lacking the influence of internal parts. Additionally, the interior is complicated further by the crack depth, carrier distribution location and water–oxygen conditions. Therefore, the method cannot truly measure the variation in repair efficiency.

The crack water flux is the amount of water that passes through the penetration path per unit of time and can reflect the actual damage within the concrete. Water flux differences were used to describe changes in crack surfaces and internal repair properties. The correlation between the microbial activation, water–oxygen environment and time effects was constructed to model the temporal variations in the cracking of microbial self-healing materials.

### 4.2. Time-Dependent Model for Repair after Cracking

Combined with the foundation of the previous study, the time-dependent model of microbial SHC repair based on RA carriers considered four main factors: bacterial solution concentration, carrier substitution rate, repair time and temperature. Assuming that the ionic reaction time conditions are neglected and the temperature is set to the peak microbial activity temperature (water temperature) of 25 °C, the variation in crack repair width is influenced by two factors, including the concentration of the bacterial solution *C_b_* and carrier utilization *r_RA_*.
(3)$$W_{r}\left(C_{b},r_{RA}\right)=W_{0}\left[1-Max\left(\left|C_{b}-40\%\right|,0.97\left|r_{RA}-30\%\right|\right)\right],$$

*r_RA_*—the carrier replacement ratio, which includes both *r_RCA_* (the RCA carrier) and *r_RFA_* (the RFA carrier)

The concentration of the bacterial solution and the carrier replacement ratio are influenced by the results of the orthogonal design tests, the economics of the microorganisms and the mechanical properties of the concrete. The values selected for both are recommended to be taken from the following ranges.
(4)$$\left(0\leq C_{b}\leq 40\%,0\leq r_{RCA}\leq 45\%,0\leq r_{RFA}\leq 30\%\right),$$

The crack repair width W_r_ boundary and initial conditions were set as follows.
(5)$$W_{r}\left(0,r_{RA}\right)=0,W_{r}\left(\infty ,r_{RA}\right)=W_{0}\left(r>0\right),$$
(6)$$W_{r}\left(C_{b},0\right)=0,W_{r}\left(40\%,30\%\right)=W_{0},$$

The resulting time-dependent deposition adjustment factor β was expressed as follows.
(7)$$\beta =3.57\times 10^{-3}\times t\times 0.9K_{t}\times \frac{W_{r}\left(C_{b},r_{RA}\right)}{W_{0}},\langle t_{T}=T\left(t<28\right),t_{T}=28\left(t\geq 28\right)\rangle ,$$

*β*—the deposition adjustment factor.

*t*—the repair time.

*K_t_*—the temperature coefficient.

*W_r_*—the repair’s remaining width.

*W*_0_—the initial crack width.

*C_b_*—the bacterial concentration.

*r_RA_*—the RA carrier replacement ratio.

The conditions for the deposition of CaCO_3_ molecules per unit area–time are as follows when neglecting the ionic reaction time.
(8)$$F=-D\frac{d_{C}}{d_{x}}=\alpha \left(C_{0}-C_{\varepsilon }\right),$$

*F*—the CaCO_3_ deposition per unit area–time.

*α*—the deposition rate, α = D/ε.

*D*—the diffusion coefficient of Ca^2+^ in the crack solution.

*ε*—the thickness of the diffusion layer.

*C*_0_—the concentration of Ca^2+^ in the crack solution, mol/L.

*C_ε_*—the concentration of Ca^2+^ during precipitation, mol/L.

Considering the effects of the replacement ratio, the concentration of the bacterial solution, the temperature and the time, the above equation is modified as follows.
(9)$$F^{\prime }=-\left(1-\beta \right)D\frac{d_{C}}{d_{x}}=\alpha \left(1-\beta \right)\left(C_{0}-C_{\varepsilon }\right),$$

The CaCO_3_ deposition caused by *Bacillus pasteurii* was mainly in the calcite state, and its linear rate expression equation is as follows.
(10)$$v=F^{\prime }\times V_{m},$$

*v*—the linear deposition rate of CaCO_3_

*V_m_*—the molar volume of CaCO_3_ and the molar volume of calcite is 36.93 cm^3^/mol.

Then, the remaining width of the crack repair, *W_t_*, was as follows.
(11)$$W_{t}=W_{0}-W_{r},W_{r}=v\times t,$$

Due to the small crack width, the repair surface boundary could be defined as a parallel relationship (i.e., no angular difference between the planes on both sides of the repair area). Based on the parallel plate theory, the following relationship was derived.
(12)$$q=740\times I\times W^{3}\times K_{t},$$

*q*—the water flux (L/h).

*I*—the hydraulic gradient, m/m. *I* = Δ*H*/*L*.

Δ*H*—the difference in water level elevation between two points on the isobath, m.

*L*—the horizontal distance between two points, m.

Combining Equations (7), (9)–(12), the model for CaCO_3_ deposition at cracks in SHC was as follows.
(13)$$q_{t}=740\times \frac{\Delta H}{L}\times K_{t}{\left[\sqrt[{3}]{\frac{q_{0}}{740\times \frac{\Delta H}{L}\times K_{t}}}-\frac{D}{\varepsilon }\times \left(1-3.57\times 10^{-3}\times t\times 0.9K_{t}\frac{W_{r}\left(C_{b},r_{RCA}\right)}{W_{0}}\right)\left(C_{0}-C_{\varepsilon }\right)\times V_{m}\times t\right]}^{3},$$

Simplify Equation (12) is as follows.
(14)$$q_{t}=740\times I\times K_{t}{\left[\sqrt[{3}]{\frac{q_{0}}{740\times I\times K_{t}}}-\alpha \left(1-\beta \right)\left(C_{0}-C_{\varepsilon }\right)\times V_{m}\times t\right]}^{3},$$

Therefore, based on the measured initial crack water flux *q*_0_, the water flux value *q_t_* of the crack in the specimen during the repair period *T* can be calculated.

Microbial SHC crack repair effectiveness factor α_r_ can characterize the degree of concrete repair at different times, as in Equation (15).
(15)$$\alpha_{r}=\frac{q_{t}}{q_{0}}\times 100\%,$$

## 5. Results and Discussion

### 5.1. Effect of Carrier Interface on Repair Efficiency through SEM

The SEM sample preparation process consists of the following main steps.

1. Locate the repair crack area and mark a 10 × 10 × 20 mm^3^ rectangular area;

2. Obtain the marked area cube using a concrete cutter;

3. Fix the sample with waterproof tape and soak it in anhydrous ethanol to prevent continuous hydration/repair of the sample;

4. Send the sample to the carrier table. The crack surface should face the electron microscope and gold spraying should be performed;

5. Scan and collect morphological data.

The carrier interface is the initial area where repair behavior is stimulated to cause healing action. It includes the raw aggregate (gray basalt, massive structure, the minerals include plagioclase, olivine and pyroxene), the OM and the ITZ between the raw aggregate and the OM. The interfaces that affect the time-dependent repair efficacy include the mortar and ITZ (old and new), as displayed in Figure 10. Among them, the OM and the old ITZ are used as microbial attachment areas, which affect the time-dependent repair efficiency through microbial activity. The new ITZ serves as an excitation condition transfer pathway that affects time-dependent repair efficacy through moisture and oxygen concentrations. At the early stage of crack extension to the carrier, moisture and oxygen are sufficient in the crack, and the repair efficacy is mainly determined by microbial activity. The process of microbial awakening and proliferation leading to repair expenditure at this stage forms the early time-dependent trend of repair efficacy. At the late stage of healing at the carrier towards the crack, the microbial activity reaches its peak and the repair efficacy is mainly determined by the amount of H_2_O and O_2_ transmitted by the progressively smaller crack. The time-dependent late development trend of the repair efficacy of microorganisms at this stage of the process of repair results from the lack of H_2_O and O_2_ causing the transition to dormancy again.

The RCA and RFA at the fractures were wrapped by the repair products, which were found via SEM, as shown in Figure 11. There were almost no repair products or hydration products (C-S-H) at the raw aggregate interface, which was due to the density and high strength of the raw aggregate, and it was extremely difficult for microorganisms to adhere to the exposed surface of the raw aggregate. The OM was abundantly distributed on the carrier surface, and the repair products were attached to the old mortar and extended outward. The larger the volume of the OM, the larger the particle size and total repair products, which indicates that the old mortar directly influences the distribution pattern and the total amount of CaCO_3_ produced by the attached microorganisms.

The sample group with RCA as the carrier had irregular aggregated CaCO_3_ distributions on the attached mortar and the ITZ. Among them, only a small amount of repair products appeared on the aggregate due to the dense and smooth texture, and the repair products were sporadically distributed with small particle sizes. The CaCO_3_ generated on the attached mortar, as observed through the RCA carrier profile, had the characteristics of large particle sizes and tight aggregation, which indicated that the attached mortar gave a better repair environment for microorganisms. The aggregated CaCO_3_ obviously improved the resistance to penetration at the concrete cracks, which coincides with the results of impermeability tests.

The RFA group specimens show that more adhered mortar from RFA did not make more repair products appear, while the area occupied by cementitious materials (such as C-S-H, etc.) was larger and fewer repair products adhered to the cementitious materials. This may be due to the fact that concrete cracking only had a small chance of occurring in the RFA. The RCA had more microbial volume than the RFA, which had a larger particle size and more adherent mortar monomer volume. Further hydration of the cementitious material with the RFA also inhibited the microbial repair effect, i.e., it hindered microbial contact with O_2_ and H_2_O. The lower remediation efficiency and the smaller distribution density of individual microorganisms resulted in smaller particle sizes and distribution areas for the products. It is worth noting that the areas where CaCO_3_ was generated are mostly places where less gelling material was present.

The NA interface of the specimens was observed to have only a small amount of cementitious material and no repair products on the larger aggregate area. This indicates that the smooth and dense nature of the natural aggregate (NA) is also not conducive to the adhesion of cementitious materials but effectively reduces the pores within the concrete and improves the overall impermeability. This also indicates that the loose and porous nature of the mortar to which the RA adheres contributes to the self-healing efficiency, a characteristic that is not present in concrete prepared with NA.

Contrast analysis of RA and NA interface characteristics found that the RA is rougher and old defects can be clearly observed at the ITZ between the aggregate and the repaired product, which is due to the poor process of the initial pouring of the RA (Figure 12). No or little cementitious material or repair products appeared in the area of the unattached mortar, which is consistent with the concrete prepared with NA. The surface of the NA specimens was smooth, and the aggregate boundary was free of adherent mortar and repair products. In fact, the NA was wrapped by the cementitious material, and incomplete wrapping in the larger aggregate areas led to the appearance of larger voids or defects. It should be pointed out that the voids in RA are mostly due to the micropores of the cementitious material after hydration and not attachment mortar area voids. The RA attachment mortar area is affected by self-healing behavior without obvious defects. The voids in the NA group are mostly due to the dense and smooth texture, resulting in the cementitious material after hydration not being better wrapped in NA.

### 5.2. Effect of Carrier Interface on Repair Product via XRD

The calcite state of CaCO_3_, as a repair product (SEM-observed morphology), is influenced by the carrier interface, and its particle size and generated amount have a direct impact on the effectiveness of the repair. The quantitative analysis of the physical phases at the repair cracks when the RCA and RFA were used as the carriers was performed via XRD. The calculation of the grain size requires obtaining the value of the full width at half maximum (FWHM) of the diffraction peaks. The sample is deconvoluted from it according to Equation (16) to obtain the broadened FW(S) due to grain refinement.
(16)$$FW{\left(S\right)}^{D}=FWHM^{D}-FW{\left(I\right)}^{D},$$

*D*—transposed convolution parameters, defined as a value between 1 and 2, where the diffraction peak pattern can be represented by either the Cauchy function or the Gaussian function. If the peak shape is closer to the Gaussian function, it is taken as D = 2. If it is closer to the Cauchy function, it is taken as D = 1. In this paper, the value of D is set to 2.

*FWHM*—the full width at half maximum of diffraction peaks.

*FW*(*I*)—the linear width value of the instrument.

*FW*(*S*)—the width value of the specimen.

After obtaining the specimen width value, the grain size is calculated according to the Scherrer formula (Equation (17))
(17)$$C_{size}=\frac{K\lambda }{FW\left(S\right)\times cos\left(\theta \right)},$$

*C_size_*—the size of the crystal.

*K*—the constant, taken as K = 1.

*λ*—the wavelength of X-ray, nm.

*θ*—the angle of diffraction (two-theta), rad.

As shown in Figure 13 and Figure 14, the diffraction angle 2T (two-theta) and FWHM of the CaCO_3_ in the RCA and RFA were 29.672, 0.109 and 29.317, 0.147, respectively. The products obtained from the physical phase analysis are mainly CaCO_3_, SiO_2_ and hydration products when the RCA is the carrier. Based on Equations (14) and (15), the average grain size of the repair products in RCA was calculated to be about 14.92 μm. The proportion of its physical phase is about 46.0% of the measured sample fraction, followed by SiO_2_ contained in siliceous aggregates, with a proportion of about 28.0%. The products obtained from the RFA specimen physical phase analysis are mainly CaCO_3_, SiO_2_, hydration products, a small amount of H_2_O and CO_2_, etc. The average grain size of the RFA repair products is calculated to be about 13.83 μm. The proportion of its physical phase is about 28.7% of the measured sample fraction, followed by SiO_2_ contained in siliceous aggregates, with a proportion of about 25.2%.

The analysis results showed that the RCA group generated a larger CaCO_3_ particle size than the RFA group, which indicated that the carrier interface of the RCA group enhanced the quality of the repair products more significantly. The larger particle size allowed the single-grain repair products to exhibit better resistance to deformation and better filling of the cracks. In addition, more CaCO_3_ was generated in the RCA group, about 1.6 times more than that in the RFA group, indicating that the amount of repair products was significantly enhanced. The greater level of production resulted in a faster repair speed and denser aggrege of repair products, which directly influenced the performance improvement in the repaired specimens. Small amounts of H_2_O and CO_2_ were found in the RFA specimens. On the one hand, this may be due to the higher water absorption rate, which makes the pore water in the adherent mortar migrate outward during the hydration process; on the other hand, it may be that the two were not completely absorbed during the repair process from the cracks into the concrete.

### 5.3. Verification of Time-Dependent Model for Repair after Cracking

The test values and calculated values are shown in Table 6, and after calculating the standard deviation of the qt test values, the coefficient of variation of the calculated values for each group was obtained by bringing in the coefficient of variation formula.

Coefficient of variance (CV):(18)$$C_{size}=\frac{K\lambda }{FW\left(S\right)\times cos\left(\theta \right)},$$

According to the analysis of the results, it is found that the time-dependent repair model has a higher accuracy when the calculated value is less than or equal to 0.5 mm, while the crack width of 1 mm is not applicable to the time-dependent repair model because the microorganisms basically do not appear or rarely appear in the repair products. In addition, the accuracy of the calculated value of the model decreases when the size of the carrier changes, so it is recommended to use this model. For the calculation, the carrier size of RCA should not exceed 15 mm.

The repair of each initial width at different sampling points with time was measured by a crack observer. The analysis revealed that the smaller specimens with initial cracks showed little repair behavior at 3 day. Some areas of the cracks healed from the sides to the middle, but a large number of pores still allowed water to flow into the concrete. The larger areas of the initial crack in the concrete show a more pronounced calcite state of CaCO_3_ inside and extending to the outside of the crack at 7 day. This is probably due to the fact that the larger cracks did not allow the CaCO_3_ generated on both sides to support its own weight while connecting to the middle and generating it inside the cracks. In addition, larger initial cracks allow moisture and oxygen fluxes to saturate the internal concentration faster than smaller ones, which favor the repair behavior of microorganisms, and again, larger parts also allow lower repair rates at the surface. This is detrimental to the corrosion resistance of concrete. At 14 day of repair, the filling of pores with repair material within the cracks could already be observed, but there were still a number of pores that were not filled. This improved the internal mechanical properties of the concrete to some extent, but the improvement in durability is still unfavorable due to the presence of voids. The observation of the smaller initial crack specimens at 28 day and 56 day revealed that the healed specimen generated repair material to cover and fill the crack surfaces, which effectively prevented the erosion of harmful substances into the concrete interior through the cracks. It should be noted that the initial crack width (>4 mm) in the 28 day to 90 day repair period remained unrepaired, and the internal repair was not significantly enhanced (Figure 15). This may be due to the fact that the flux of H_2_O and O_2_ into the microbial repair area was weakened while the interior was better filled, which instead lessened the repair effectiveness and meant that the repair of the cracked fecal surface could not be completed.

## 6. Conclusions

The incorporation of RAs enhanced the resourcefulness, greenness and economic properties of microbial SHC. In addition, the experimental results demonstrate the positive repair effects of both RCA and RFA on cracking. The rate of RA as a repair carrier instead of NS&G can be increased with the design demand, which is beneficial to resolving the high replacement utilization problem of RA. *Bacillus pasteurii*, as a clean repair driver, meets the current demand for sustainable building development.

(1) In the comparison among RCA, RFA and NA as carriers, the loose and porous nature of the adhered mortar of RA makes the cracks extend more along the aggregate when cracking. The higher level of adhered mortar needed for RCA makes the local microbial density more obvious than that of RFA, and the smoothness of NA is not conducive to microbial adherence, which in turn leads to lower repair efficiency.

(2) The products of the specimens with RCA and RFA as carriers that were obtained in the physical phase analysis were mainly CaCO_3_, SiO_2_ and hydration products. The average particle size of the RCA group was calculated to be about 14.92 μm, according to the formula. The proportion of its physical phase accounted for about 46.0% of the measured sample fraction. In contrast, the average particle size of the repair products in the physical phase analysis of the RFA group was about 13.83 μm, which was 92.69% of the RCA group; the proportion of its physical phase was about 28.7% of the measured sample fraction, which was 62.39% of the total of the RCA group. This indicates that the RCA group is better than the RFA group in terms of particle size and the total amount of repair products.

(3) Both recycled coarse and fine aggregates as carriers enhanced the concrete crack repair rate, compared to the SC and NC groups. At 14 days of repair, the remaining crack, with an initial width of 0.15 mm, was 0.06 mm in the SCC group, thus 60% repaired, while the remaining crack width in the SFC group was 0.11 mm (only 26.67% repaired). The crack repair was difficult to achieve by microbial action alone; only 13.33% and 10% of the cracks with initial 0.15 mm and 0.2 mm widths were repaired in the SC group, respectively. The larger cracks continued to extend compared to the initial cracks, which aggravated the deterioration of the properties.

## Figures and Tables

**Figure 1 materials-16-02371-f001:**
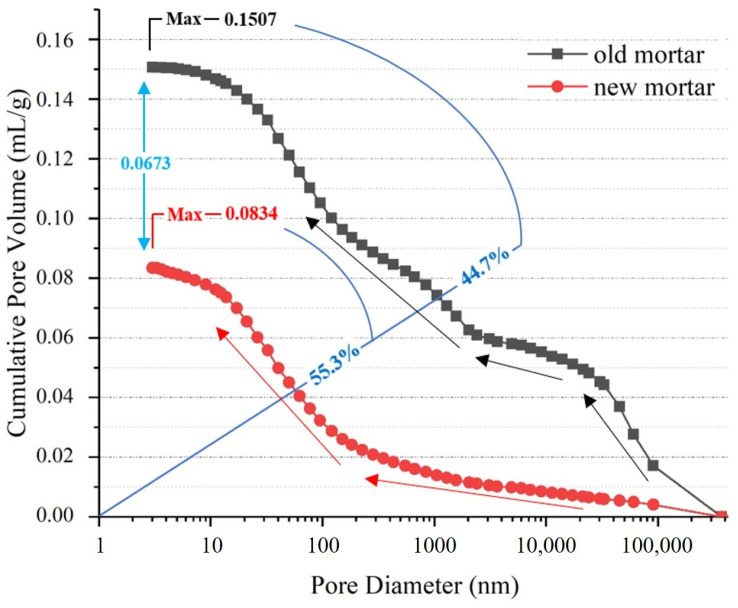
Trend of pore size and accumulated pore volume of OM and NM. Note: the mercury intrusion process is from the positive direction of the X-axis to the origin.

**Figure 2 materials-16-02371-f002:**
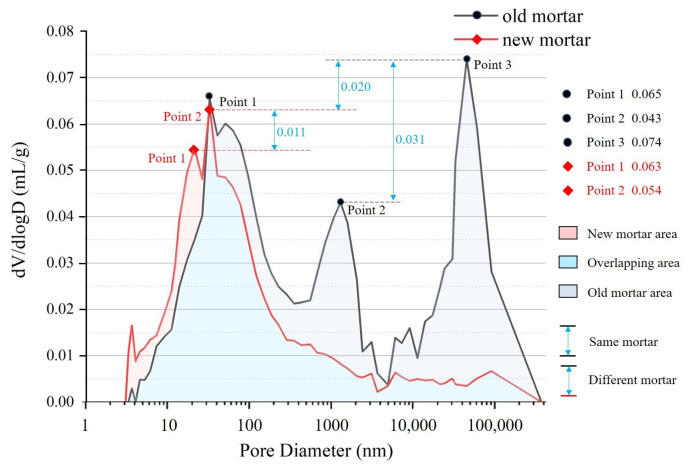
Trends of *dV*/*dlogD* pore volume and pore size for OM and NM.

**Figure 3 materials-16-02371-f003:**
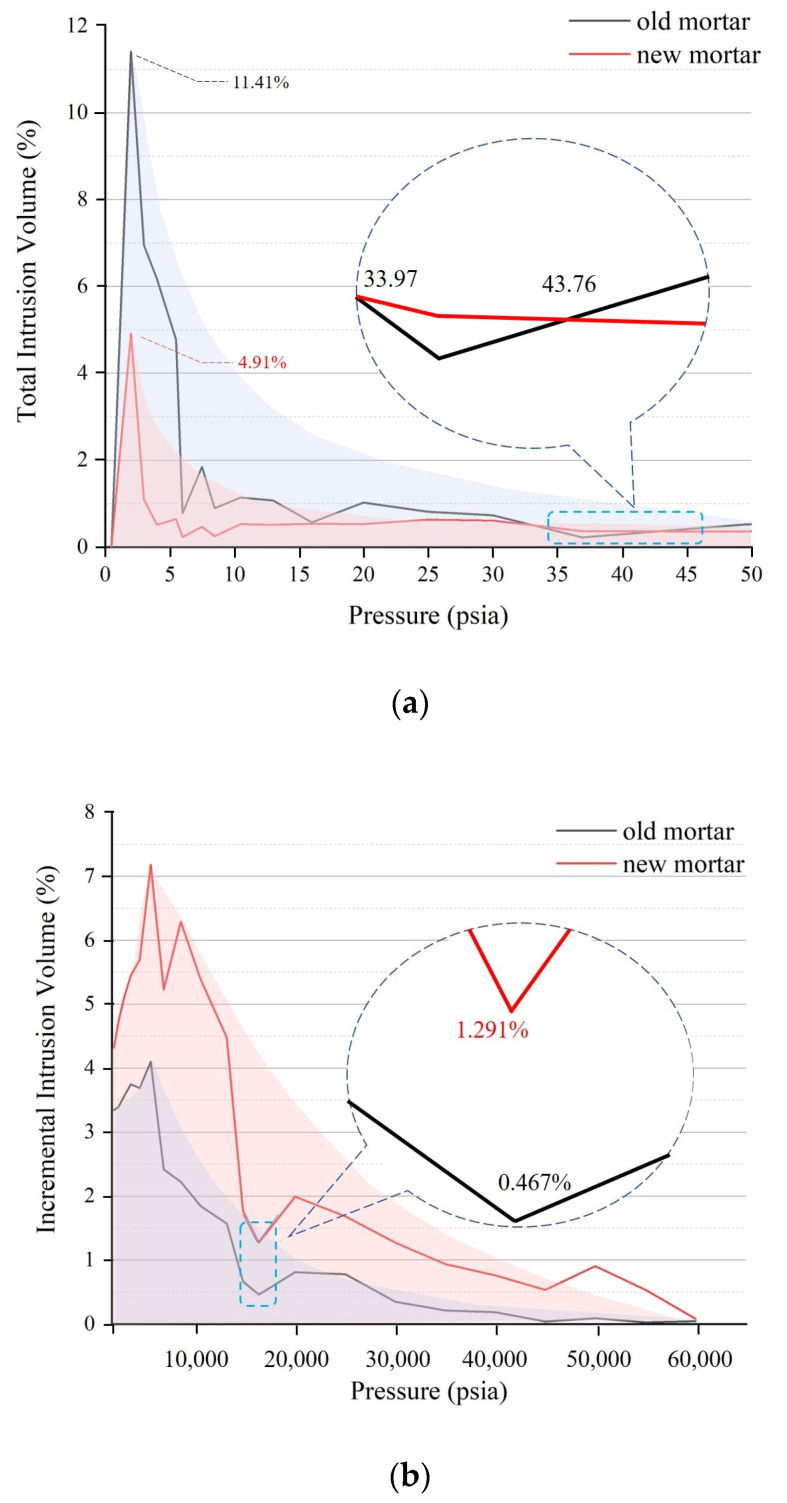
Trend of increasing pressure and intrusion of OM and NM: (**a**) early stage of MIP process; (**b**) end stage of MIP process.

**Figure 4 materials-16-02371-f004:**
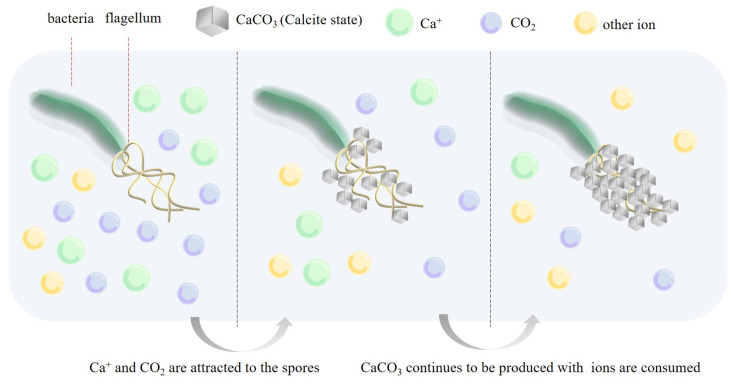
Relationship between repair behavior and ion migration within cracks.

**Figure 5 materials-16-02371-f005:**
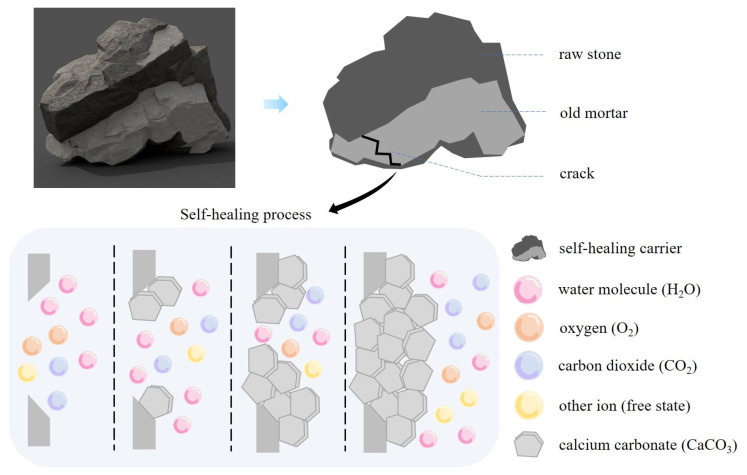
Schematic representation of the activation of self-healing to crack repair.

**Figure 6 materials-16-02371-f006:**
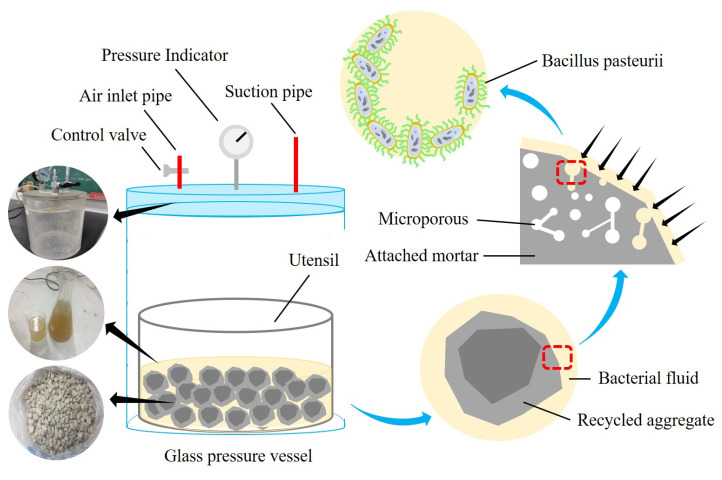
Schematic representation of microbial adsorption on the carrier.

**Figure 7 materials-16-02371-f007:**
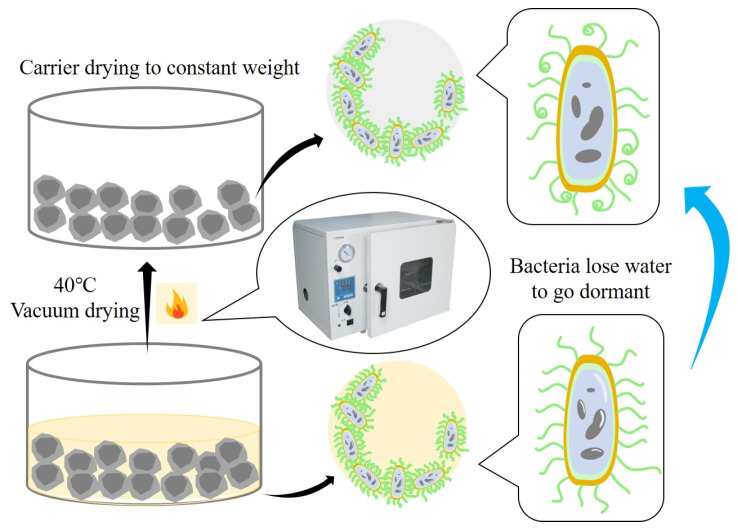
Schematic representation of carrier drying after adsorption.

**Figure 8 materials-16-02371-f008:**
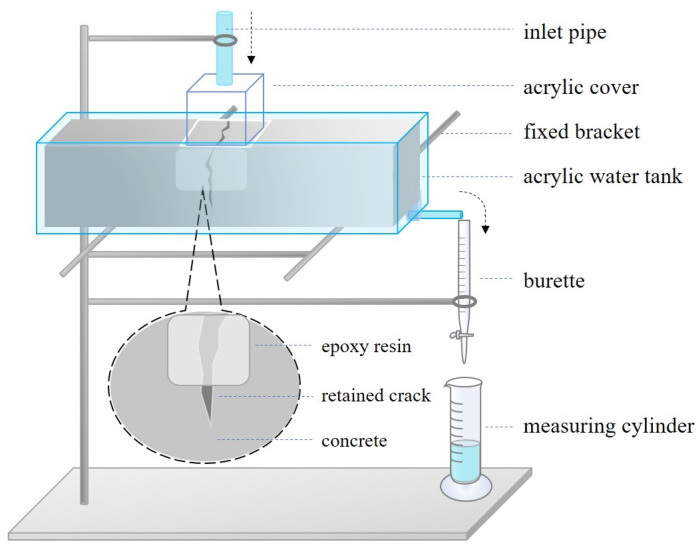
Schematic diagram of the specimen water penetration test device.

**Figure 9 materials-16-02371-f009:**
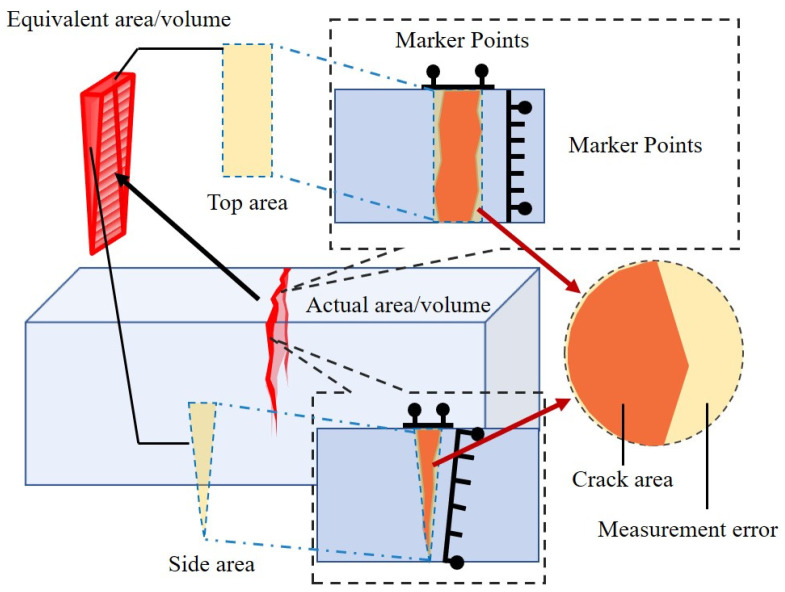
Schematic representation of specimen repair markings and calculations.

**Figure 10 materials-16-02371-f010:**
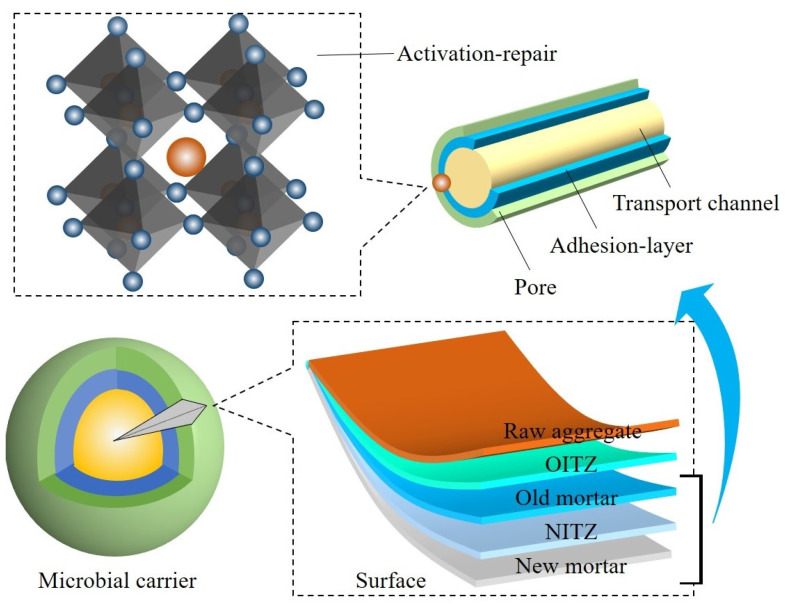
Schematic representation of the microbial carrier interface to generate products.

**Figure 11 materials-16-02371-f011:**
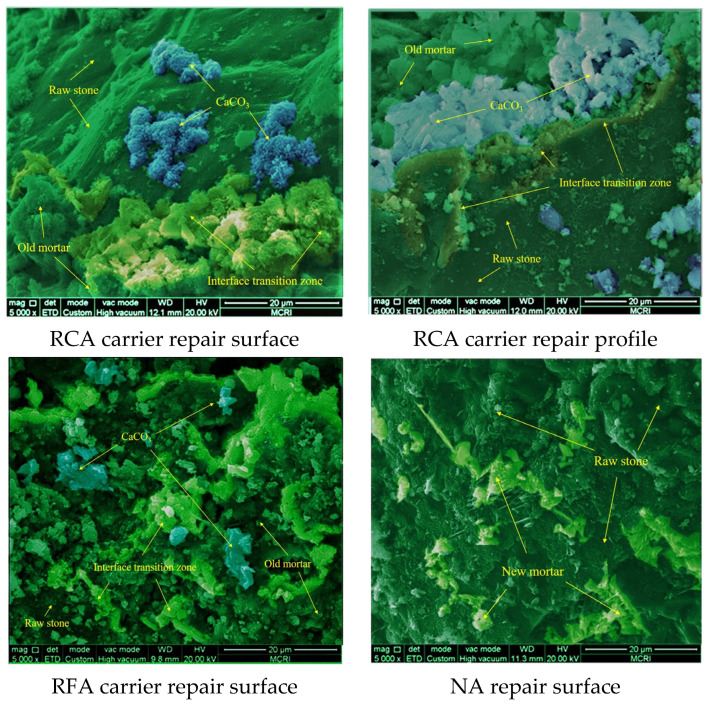
The repair morphology of carrier interface.

**Figure 12 materials-16-02371-f012:**
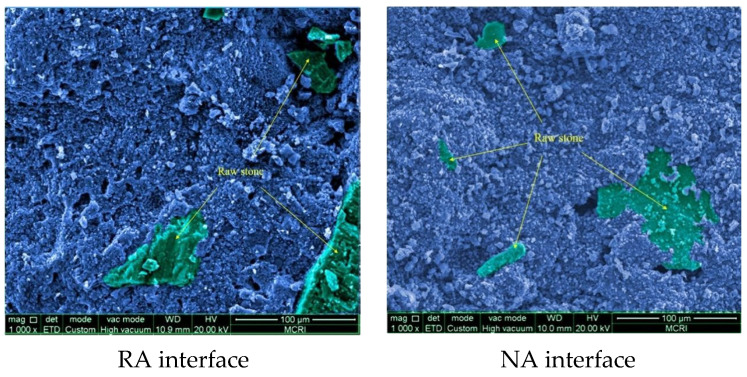
Comparison of repair morphology of different aggregate interfaces.

**Figure 13 materials-16-02371-f013:**
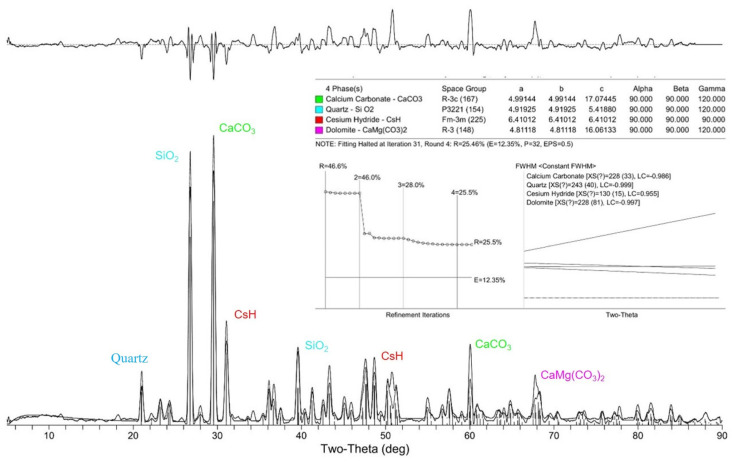
Physical phase analysis of RCA group specimens.

**Figure 14 materials-16-02371-f014:**
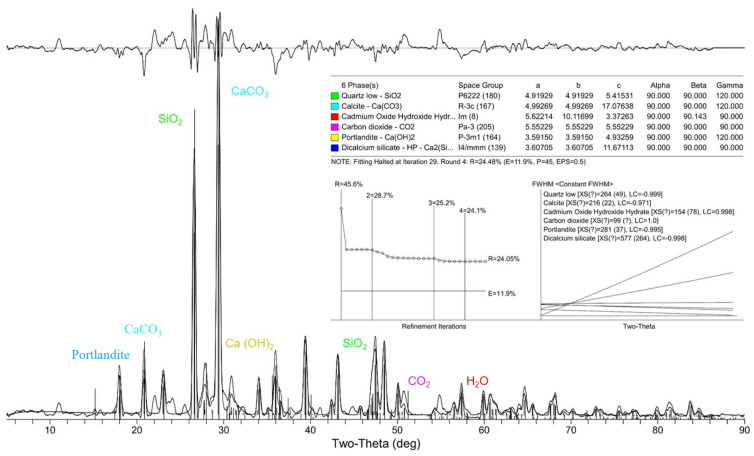
Physical phase analysis of RFA group specimens.

**Figure 15 materials-16-02371-f015:**
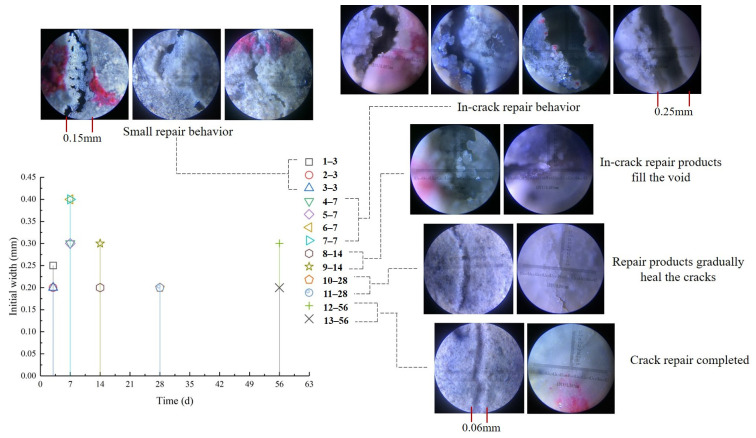
Physical phase analysis of RFA group specimens (Specimen X-Y, X is the marker point number, Y is the measured repair time).

**Table 1 materials-16-02371-t001:** Bacterial strain medium composition.

UPW/L	Peptone/g	Meat Extracts/g	NaHCO_3_/g	Na_2_CO_2_/g	Agar/g
1	5	3	0.42	0.53	15

Note: 5 mg MnSO_4_-H_2_O was added during the cultivation of *Bacillus pasteurii*, which facilitated the production of spores. The conventional culture time of the bacterial strain is 1–2 days when the suitable pH value is 7.2–8.0 and the fermented condition is 34 °C.

**Table 2 materials-16-02371-t002:** Physical properties of recycled aggregate carriers.

Carrier Type	Apparent Density/kg·m^−3^	Crushing Value/%	Moisture Content/%	Water Absorption/%	Void Content/%	Porosity/%	Average Pore Size/nm
RCA	2458	17.0	1.33	3.83	29.5	28.63	782.1
RFA	2390	24.3	4.15	12.30	43.0	31.32	65.7

**Table 3 materials-16-02371-t003:** Chemical composition of cement.

Component	SiO_2_	Al_2_O_3_	Fe_2_O_3_	CaO	TiO_2_	MgO	SO_3_	Na_2_O	K_2_O	P_2_O_5_	Others
Content	21.1	4.3	2.5	65.9	0.1	1.5	2.3	0.1	0.5	0	1.7

**Table 4 materials-16-02371-t004:** Specimens’ mix ratio.

Specimen No.	RCA /kg	NCA /kg	RFA /kg	Sand /kg	Cement /kg	Mixing Water/kg	Additional Water/kg	Concentration of Bacterial Solution/kg	Calcium Lactate/kg
SCC	338.4	828.0	0	647.0	384.0	183.0	8.4	74.6(D)	10.2
SFC	0	1186.0	189.2	453.0	384.0	183.0	15.4	74.6(D)	10.2
SC	0	1186.0	0	647.0	384.0	108.4	0	74.6(D)	10.2
NC	0	1186.0	0	647.0	384.0	183.0	0	0	10.2

**Table 5 materials-16-02371-t005:** Specimen pre-set crack width calibration details/(mm).

Specimen No.	Average Value			Maximum Value	Minimum Value	Number of Marker Points
	Point 1	Point 2	Point 3			
SCC-1	0.24	0.16	0.04	0.25	0.03	18
SCC-2	0.22	0.08	0.02	0.23	0.02	18
SCC-3	0.26	0.10	0.05	0.27	0.03	18
SFC-1	0.23	0.09	0.01	0.24	0.01	18
SFC-2	0.20	0.07	0.03	0.22	0.02	18
SFC-3	0.24	0.11	0.05	0.26	0.03	18
SC-1	0.22	0.13	0.04	0.22	0.02	9
SC-2	0.20	0.05	0.01	0.20	0.01	9
SC-3	0.22	0.09	0.02	0.24	0.01	9
NC-1	0.26	0.07	0.04	0.27	0.03	9
NC-2	0.19	0.11	0.02	0.21	0.01	9
NC-3	0.22	0.08	0.03	0.24	0.03	9

**Table 6 materials-16-02371-t006:** Comparison between calculated and tested values of water flux of repaired 14-day specimens/(L/h).

Specimen No.	*q* _0_	*q_t_* (Test Value)	*q_t_* (Calculated Value)	W_0_	W_r_	ε/mm	α/10^−5^	Coefficient of Variance (CV)
SCC-1	0.324	0.207	0.196	0.15	0.06	15	1.20	2.80%
SCC-2	1.117	0.974	0.811	0.2	0.14	13	1.38	16.73%
SCC-3	4.404	4.277	4.069	0.3	0.28	21	0.86	4.87%
SFC-1	0.576	0.479	0.453	0.15	0.11	16	1.13	5.43%
SFC-2	0.624	0.580	0.513	0.2	0.17	18	1.00	11.55%
SFC-3	1.768	1.752	1.690	0.3	0.29	18	1.00	3.54%
SC-1	0.732	0.686	0.669	0.15	0.13	13	1.38	2.48%
SC-2	0.351	0.348	0.303	0.2	0.19	12	1.50	12.93%
SC-3	1.986	2.216	1.445	0.3	0.32	18	1.00	34.79%
NC-1	0.228	0.229	-	0.15	0.15	14	-	-
NC-2	1.435	1.411	-	0.2	0.2	16	--	-
NC-3	0.662	0.894	-	0.3	0.3	16	-	-

## Data Availability

The data presented in this study are available on request from the corresponding author. The data are not publicly available due to continuity of long-term analytical tests.

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
