# Peer review of "Study on the Performance of Recycled Coarse and Fine Aggregates as Microbial Carriers Applied to Self-Healing Concrete"

_materials, 2023, doi:10.3390/ma16062371_

Round 1

Reviewer 1 Report

In relation to the work "Study on the performance of recycled coarse and fine aggregates as microbial carriers applied to self-healing concrete" the following should be considered:

The summary should be reinforced with the data or values that were obtained to provide greater interest to your work. Likewise, give relevance to the objective, and the hypothesis of its study.

It is necessary to improve the quality of the figures and adjust the scale.

XRD samples need to be better indexed, perhaps as indicated by the technique with a minimum of four peaks for phase identification, and include PDFs and crystallographic planes. (Figures 14-15). Likewise, consider FWHM – the full width at half maximum of diffraction peaks. But each species

The author should elaborate on the discussion of how it affects or benefits pozzolanity in concrete.

Add more current bibliographical citations

Reviewer 2 Report

The authors proposed the utilization of recycled aggregates as microbial carriers in microbial self-healing concrete. Both recycled coarse aggregate and recycled fine aggregate were used. The results showed that the efficiency of the self-healing process was affected by adhered old mortar. The manuscript is very well organized, and easy to follow. However, some information in the manuscript needs to be clarified.

1) Line 64: Please insert “coarse” together with natural aggregate, since both gravel and sand are aggregates.

2) Figure 1 does not provide additional information to the text. I suggest removing it.

3) Some figures present low resolution and small font size. Please, increase the figures.

4) You cannot use Figure 5 in the manuscript since it has already been published elsewhere. In this case, a citation is required along with permission to use published figures.

5) Section 3.2 Mixing ratio design: Why is the reason of 30% carrier replacement? Was this content chosen in order to avoid a decrease in mortar performance caused by the use of recycled aggregates? To improve the self-healing process efficiency? Or a combination of both?

6) Section 3.2 Mixing ratio design: “Based on the previous experimental work [39], 54 standard specimens of 40*40*160 mm3 (30% carrier replacement ratio, 5-10 mm RCA particle size; 1.18- 4.75 mm RFA particle size, 40% concentration of bacterial solution, and 2 mm precast crack width) were prepared.”

Does the 30% content refer to the replacement of natural aggregates by recycled aggregates? The word “carrier” is suggesting that natural aggregates are also carrier. Make it clearer in the text.

7) Line 245: There is a typo. Please replace “;” with “.”.

8) Line 281: There is a typo since Table 3 is chemical composition.

9) Describe the sample preparation process for SEM.

10) As shown in Figures 11, 12 and 13, a secondary electron detector was used. How was the separation of the compounds performed? How was the separation of old mortar and cement paste carried out? If the analyzes were done by morphology, use high-magnification images that illustrate the typical characteristics of each compound.

11) Lines 492-494. “The average grain size of the RFA repair products is calculated to be about 13.83 μm.” This sentence was written twice.

12) Were the samples for XRD powder? In this case, was the entire sample ground or just the cement paste?

13) Recycled aggregates were provide from demolished ring beams and structural columns. What type of aggregate (stone and minerals) is used in this structure?

14) Figures 14 and 15: Diffraction peak at 2θ ~ 20.88 belongs to quartz while 2θ ~ 18.08 is Portladite. In that region, calcite presents a low intensity peak at around 23.09. Please correct the identification of phases in the diffractogram.  

15) The amount of CH can also be a way of identifying whether the self-healing process is occurring. The diffractograms suggest that there may be less CH in the RCA sample than in the RFA sample.

16) It is suggested that the authors link the results observed for different properties considered in this study with more bibliographic references. The discussion section must be further improved.

Reviewer 3 Report

The authors demonstrated the microbial self-healing concrete with combined recycled coarse and fine aggregates (RCA and RFA) as microbial carriers. Unlike the previous literature (Ref. 39), the authors investigated the effect of the mixtures of RCA and RFA on the self-healing performance and developed the time-dependent repair model. It will be beneficial of the development of recycled concrete. However, the reviewer can recommend the article to be published in this journal after the authors address the following comments:

  1. Which condition is more effective to heal the cracks? Also, Which RA is more useful for microbial self-healing concrete? Please compare the result of the previous one reported in Ref. 39.
  2. The authors should provide the values of rRA investigated in this study.
  3. The authors should provide the supporting results to confirm materials’ components in SEM images.
  4. Please provide the schematic illustration of self-healing mechanism studied in this work.
  5. Another method to analyze the self-healing efficiency (compressive test before and after healed) should be added, which will be necessary in these kinds of fields.
  6. The letters’ size in Figures and Tables should be tuned for the readers.
  7. The English written in this manuscript should be checked.

Reviewer 4 Report

The authors explore the function of recycled coarse and fine aggregates as carriers on the repair performance of microbial self-healing concrete. Overall, the study is relatively novel, and the findings are interesting and will be useful. However, they must consider the following comments before this can be published in MDPI Materials.

(1)  The authors show SEM micrographs in figures 12 and 13. They color the micrographs with different colors like green, blue etc. and have made conclusions about which structure represents what. What is the basis of their structure assignment? Only secondary electron contrast? Even low Z materials that do not conduct, can appear bright due to charging. Did they perform EDS or elemental mapping? Also, apart from these colored micrographs they must provide the original grayscale images for the readers. They could also perform SEM at low keV. 20keV is very high and can damage any beam sensitive specimen that might be present.

(2)  What temperature are they measuring the XRD at? Are they performing experiments under cryo conditions? If not, why do they see peaks for H2O/ Water. Crystalline Ice can diffract and as a result can show sharp peaks, not melted water. They need to provide some explanation on this. 

(3)  In general, the figures must be described in relatively more detail. The current captions are short and need more details about the panels/ figures. Also, the fonts of labels in figure 10 is small. Need to increase font size.

(4)  Add scale bars in figure 16.

Round 2

Reviewer 2 Report

Thank you for responding to all my comments.
I suggest that the review information be included in the manuscript. This information is important for readers. For example, I asked about the mineralogy of the aggregates because if it is limestone, it could affect the analysis of the results because CaCO3 is produced by the self-healing process.
The same applies to sample preparation and analysis of SEM.
